# Mass Spectrometry as a Highly Sensitive Method for Specific Circulating Tumor DNA Analysis in NSCLC: A Comparison Study

**DOI:** 10.3390/cancers12103002

**Published:** 2020-10-16

**Authors:** Pierre-Jean Lamy, Paul van der Leest, Nicolas Lozano, Catherine Becht, Frédérique Duboeuf, Harry J. M. Groen, Werner Hilgers, Nicolas Pourel, Naomi Rifaela, Ed Schuuring, Catherine Alix-Panabières

**Affiliations:** 1Biopathologie et Génétique des Cancers, Institute d’Analyse Médicale Imagenome, Inovie, 6 Rue Fontenille, 34000 Montpellier, France; nicolas.lozano@labosud.fr; 2Department of Pathology, University Medical Center Groningen, University of Groningen, Hanzeplein 1, 9713 GZ Groningen, The Netherlands; p.van.der.leest@umcg.nl (P.v.d.L.); n.b.rifaela@umcg.nl (N.R.); e.schuuring@umcg.nl (E.S.); 3Oncologie Médicale, Clinique Clémenville, 25 rue Clémenville, 34000 Montpellier, France; catherine.becht@laposte.net (C.B.); frederique.duboeuf@gmail.com (F.D.); 4Department of Pulmonary Medicine, University Medical Center Groningen, University of Groningen, Hanzeplein 1, 9713 GZ Groningen, The Netherlands; h.j.m.groen@umcg.nl; 5Oncologie Médicale, Institute Sainte Catherine, 250 Chemin de Baigne Pieds, 84918 Avignon, France; w.hilgers@isc84.org (W.H.); n.pourel@isc84.org (N.P.); 6Laboratoire de Cellules Rares Circulantes, University Medical Center of Montpellier, 641, Avenue du Doyen Gaston GIRAUD, 34093 Montpellier, France; c-panabieres@chu-montpellier.fr

**Keywords:** non-small cell lung cancer, EGFR mutation, circulating DNA, tyrosine kinase inhibitors, liquid biopsy

## Abstract

**Simple Summary:**

We compared the UltraSEEK™ Lung Panel on the MassARRAY^®^ System (Agena Bioscience) with the FDA-approved Cobas^®^ EGFR Mutation Test v2 for the detection of *EGFR* mutations in liquid biopsies of NSCLC patients, accompanied with preanalytical sample assessment using the novel Liquid IQ^®^ Panel. For the detection of relevant predictive mutations using the UltraSEEK™ Lung Panel, an input of over 10 ng showed 100% concordance with Cobas^®^ EGFR Mutation Test v2 and detection of all tissue confirmed mutations. In case of lower ccfDNA input, the risk of missing clinically relevant mutations should be considered. The use of a preanalytical ccfDNA quality control assay such as the Liquid IQ^®^ Panel is recommended to confidently interpret results, avoiding bias induced by non-specific genomic DNA and low input of specific tumoral ccfDNA fragments.

**Abstract:**

Plasma-based tumor mutational profiling is arising as a reliable approach to detect primary and therapy-induced resistance mutations required for accurate treatment decision making. Here, we compared the FDA-approved Cobas^®^ EGFR Mutation Test v2 with the UltraSEEK™ Lung Panel on the MassARRAY^®^ System on detection of *EGFR* mutations, accompanied with preanalytical sample assessment using the novel Liquid IQ^®^ Panel. 137 cancer patient-derived cell-free plasma samples were analyzed with the Cobas^®^ and UltraSEEK™ tests. Liquid IQ^®^ analysis was initially validated (*n* = 84) and used to determine ccfDNA input for all samples. Subsequently, Liquid IQ^®^ results were applied to harmonize ccfDNA input for the Cobas^®^ and UltraSEEK™ tests for 63 NSCLC patients. The overall concordance between the Cobas^®^ and UltraSEEK™ tests was 86%. The Cobas^®^ test detected more *EGFR* exon19 deletions and L858R mutations, while the UltraSEEK™ test detected more T790M mutations. A 100% concordance in both the clinical (*n* = 137) and harmonized (*n* = 63) cohorts was observed when >10 ng of ccfDNA was used as determined by the Liquid IQ^®^ Panel. The Cobas^®^ and UltraSEEK™ tests showed similar sensitivity in *EGFR* mutation detection, particularly when ccfDNA input was sufficient. It is recommended to preanalytically determine the ccfDNA concentration accurately to ensure sufficient input for reliable interpretation and treatment decision making.

## 1. Introduction

Mutations and gene-fusions underlie key molecular mechanisms that drive cancer development and progression. Therefore, treatment strategies that target specific molecules related to gene mutations or gene-fusions have been developed. In non-small cell lung cancer (NSCLC), activating mutations of *EGFR* in exons 18–21 are well established as predictive biomarkers for treatment of patients with EGFR tyrosine kinase inhibitors (TKIs) [1]. Personalized treatment strategies for genetically stratified NSCLC subgroups improve patient outcomes, and molecular testing recommendations for treatment with targeted TKIs have been reported [2,3]. Despite the high response rates to various first- and second-generation EGFR-TKIs, eventually all patients with metastasized NSCLC with an *EGFR* mutation will develop disease progression due to acquired resistance, mostly attributable to the *EGFR* T790M mutation [4]. Osimertinib was introduced as a third-generation EGFR-TKI that selectively and irreversibly targets the *EGFR* T790M mutation [5]. In 2015, osimertinib was approved for treatment of *EGFR* T790M-positive patients who have progressed on first- or second- generation EGFR-TKIs [6]. In addition to *EGFR* T790M, other resistance mutations in *EGFR* emerged, e.g., C797S, L718Q, S768I and exon20 insertions [7]. Other mutated genes such as *KRAS*, *BRAF*, *PIK3CA* and *ERBB2* have been reported in patients treated with first- or second-generation and more recently third-generation EGFR inhibitors [4,8]. 

*EGFR* mutation detection is nowadays performed with polymerase chain reaction (PCR)–based methods that are more sensitive than traditional Sanger sequencing and is currently implemented in many laboratories. As genetic mechanisms of secondary resistance arise in a sub-clonal manner, resulting in i.e., low-level allelic frequencies, secondary *EGFR* resistance mutation detection is even more challenging [9,10]. The development of resistance mutations and subsequent new treatment options led to the necessity of evolutive molecular tumor profiling not only at time of diagnosis but also at disease progression of patients receiving targeted therapy. However, of at least 20% of patients, clinicians are unable to obtain tumor tissue biopsy or evaluable tissue biopsy [11]. To solve this issue, liquid biopsy approaches have been introduced analyzing circulating tumor DNA (ctDNA) from cell-free plasma. Several studies support the use of ctDNA to determine the *EGFR* mutational status in cases where tumor tissue is limited or insufficient [12]. 

In cancer patients, only a fraction of the circulating cell-free DNA (ccfDNA) carries tumor-specific somatic mutations (i.e., the ctDNA fraction) against a background of circulating DNA originating from peripheral blood cells and other tissues [13]. Thus, the ctDNA fraction might account for less than 1% in many plasmas, and therefore it requires highly sensitive detection methods [13,14]. A major drawback in the implementation of liquid biopsy approaches is that preanalytical and analytical factors have not been harmonized while their effects on performance and results of ctDNA assays are major [13,14,15,16,17]. Variation of white blood cell (WBC)-derived DNA in cell-free plasma due to hemolysis during the collection, transport and processing of blood is an essential factor that can affect PCR and sequencing results by dilution of ctDNA with wild-type or non-cancerous DNA, increasing false-negative results, and erode reliable quantification of the variant allelic frequency (VAF). 

To date, only one of the very few FDA-approved ccfDNA-based tests with clinical utility is the Cobas^®^ EGFR Mutation Test v2 (Roche Molecular Systems Inc., Pleasanton, CA, USA) detecting 42 *EGFR* hotspot mutations in ccfDNA from patients with lung cancer [5,18,19,20,21,22]. The Cobas^®^ test is approved as a companion diagnostic test on cell-free plasma to select NSCLC patients eligible for EGFR-TKI, including osimertinib [23]. 

In this study, we compared the variant detection in 137 plasma samples of patients with NSCLC using the UltraSEEK™ Lung Panel on the MassARRAY^®^ System (Agena Bioscience, San Diego, CA, USA), a new highly-sensitive method to detect 67 mutations across 5 genes (*EGFR*, *BRAF*, *KRAS*, *ERBB2* and *PIK3CA*) in a one PCR multiplex assay, with the standard method Cobas^®^ EGFR Mutation Test v2. We also introduce the Liquid IQ^®^ Panel on the MassARRAY^®^ System, which provides information on the preanalytical conditions of each sample through the total input amount of amplifiable ccfDNA in a reaction, the ccfDNA size (presence or absence of high molecular weight DNA) and the level of WBC contamination, and it authenticates samples using a comprehensive set of single nucleotide polymorphisms (SNPs). We then studied the optimal input of ccfDNA determined with the Liquid IQ^®^ and its implication on tests results. 

## 2. Results

### 2.1. Comparison of Mutation Detection Using UltraSEEK™ and Cobas^®^

Plasma-derived ccfDNA of 137 patients was tested for mutation-harboring ctDNA using the UltraSEEK™ Lung Panel on the MassARRAY^®^ System and compared to the results from the diagnostic reports of the same plasma samples tested with the Cobas^®^ EGFR Mutation Test v2, according to the conditions as recommended by the manufacturers. For UltraSEEK™ analysis, the number of amplifiable copies as determined by Liquid IQ^®^ analysis were used to calculate the ccfDNA input for each sample. Consequently, the ccfDNA input for the UltraSEEK™ and Cobas^®^ analyses were not equal. Concordant detection of *EGFR* mutations by both panels was found in 118 cases (86%; Figure 1A). In 10 cases (7.3%) the Cobas^®^ detected *EGFR* mutations missed with UltraSEEK™, while the UltraSEEK™ detected mutations in 9 cases missed by Cobas^®^ (6.6%; Appendix A). Cohen’s κ revealed a substantial agreement of 0.71 (Appendix A). Considering all clinically relevant mutations covered by the UltraSEEK™ Panel (Appendix A), additional mutations were detected in 18.2% of all patients not detectable with the Cobas^®^ test, compared to 8.0% detected with the Cobas^®^ test only (Figure 1B; Appendix A). Accordingly, the agreement between UltraSEEK™ and Cobas^®^ drops to a moderate level of 0.54 (Appendix A).

A total of 115 *EGFR* mutations covered by both tests were detected with both the UltraSEEK™ and Cobas^®^ panels (Appendix A; Figure 1C). Using UltraSEEK™, more EGFR exon19 deletions and L858R mutations were missed compared with Cobas^®^ (9 versus 2, respectively). On the other hand, the Cobas^®^ missed the TKI-resistant T790M mutation in 6 out of 34 cases, while UltraSEEK™ detected all (Figure 1D; Appendix A). 

The UltraSEEK™ analysis revealed 20 additional mutations, partially other than *EGFR* mutations, and those were not covered by the Cobas^®^ Panel (Figure 1C; Appendix A). Among these mutations were common *KRAS* G12/13 mutations, as well as *EGFR* C797S and *BRAF* V600E mutations frequently associated with treatment resistance (Figure 1D; Appendix A).

### 2.2. Validation of the Liquid IQ^®^ Panel

The Liquid IQ^®^ Panel on the MassARRAY^®^ System is a novel approach to quantify and qualify the extracted ccfDNA prior to UltraSEEK™ analyses. In order to validate its applicability, 84 randomly-selected patient-derived ccfDNA samples extracted from plasma collected from the same patient simultaneously in either Streck (*n* = 42) or EDTA (*n* = 42) blood collection tubes were analyzed using the Liquid IQ^®^ Panel. Similar numbers of amplifiable copies and yield were determined between Streck and EDTA samples (Table 1; Appendix A). Despite that Qubit™ measures higher concentrations of ccfDNA (Appendix A), a strong correlation between Liquid IQ^®^ and Qubit™ was found (R^2^ = 0.87; Figure 2).

### 2.3. Quantitative and Qualitative Analysis Using the Liquid IQ^®^ Panel

All 137 clinical plasma samples were analyzed with the Liquid IQ^®^ Panel to calculate the input of amplifiable ccfDNA for each UltraSEEK™ analysis. A broad range in input amounts of 2.3 to 25 ng ccfDNA was detected (Table 2; Appendix A). The Liquid IQ^®^ analysis was performed retrospectively in the Montpellier (MTP; cases MTP001-MTP100) cohort and revealed that a relatively high number of Liquid IQ^®^ unevaluable samples were included in this cohort. When the ccfDNA concentrations determined with Liquid IQ^®^ were compared with the the LabChip^®^ (MTP cohort) measurements, a strong correlation was observed (R^2^ = 0.80; Figure 3A). For the Groningen (GRO; cases GRO01-GRO-37) cohort, a strong correlation between Liquid IQ^®^ and Qubit™ was also found (R^2^ = 0.82; Figure 3B), similar to that observed in the pilot Liquid IQ^®^ cohort (R^2^ = 0.82; Figure 2). Where Liquid IQ^®^ and LabChip^®^ on average measure similar comparable amounts of ccfDNA (0.21 ng/µL versus 0.25 ng/µL, respectively), quantification using Qubit™ measures considerably more ccfDNA than Liquid IQ^®^ (0.91 ng/µL versus 0.60 ng/µL, respectively; Appendix A).

When comparing the UltraSEEK™ and Cobas^®^ results in relation to the amount of ccfDNA input, we observed a 100% concordance in samples with an input of more than 10 ng as determined by Liquid IQ^®^. A minimum input of 10 ng (i.e., 3000 copies equivalent genome) is recommended to achieve a sensitivity of 0.1% (3 copies equivalent genome) for UltraSEEK™ analysis [24] and was affirmed for the detection of *EGFR* mutations present on both panels (Table 3). With an input of 8–10 ng ccfDNA, the concordance drops to 90%. Taken together, the concordance of UltraSEEK™ and Cobas^®^ using a (sub-)optimal input of >8 ng for UltraSEEK™ is 94%. When lower ccfDNA input amounts are used, the concordance drops to 79% and 84% for 2–8 ng and Liquid IQ^®^ unevaluable samples, respectively, equally for both UltraSEEK™ and Cobas^®^ testing (Table 3). Regarding the concordance of mutation detection between UltraSEEK™ and Cobas^®^ when excluding the mutation-negative (wildtype) plasma samples (*n* = 46), a similar concordance of 93% is found for samples with a (sub-)optimal ccfDNA input (>8 ng; Table 3; Appendix A). However, for samples with lower or unevaluable ccfDNA input amounts, the concordance significantly drops to 69% and 67%, respectively (Table 3; Appendix A).

### 2.4. Comparison of UltraSEEK™ and Cobas^®^ with Harmonized Input

The Cobas^®^ test is based on a ccfDNA input of a 75% equivalent of at least 2 mL plasma (3 × 25 µL of the total extracted ccfDNA). CcfDNA input volumes for UltraSEEK™ analysis were based on the concentration of the eluate to ensure preanalytical conditions with a maximum volume of 40 µL. Due to considerable variations in plasma ccfDNA concentrations [25] (varying from 0.087–19.4 ng/µL; based on quantification with either LabChip^®^ or Qubit™), reaching (sub-)optimal levels of ccfDNA input (>8 ng) was not achievable in many cases since large sample volumes (>40 µL) cannot be loaded. Accordingly, the ccfDNA input amounts as determined by Liquid IQ^®^ analysis varied significantly (Appendix A). In order to evaluate the effect in mutation detection with the Cobas^®^ EGFR Mutation Test v2 when harmonizing the input volumes of the same eluate to equal amounts when available, we repeated the UltraSEEK™ and Cobas^®^ analyses in a subpopulation of 63 ccfDNA samples from the MTP cohort, using (sub-)optimal (*n* = 23) and lower ccfDNA input amounts (*n* = 40). 

Concordance of *EGFR* mutations detection present on both panels was observed in 78% (*n* = 49; Figure 4A; Appendix A). In contrast to the previous analysis, the UltraSEEK™ assay detected more mutations in nine cases (14%) while the Cobas^®^ test detected more mutations in only one case (2%). The agreement as determined by Cohen’s κ dropped to 0.66 (Appendix A). Furthermore, four of the Cobas^®^ analyses failed supposedly due to insufficient input (Figure 4A). UltraSEEK™ analysis revealed more clinically relevant mutations in almost a quarter of the studied patients (Figure 4B). In total, 37 mutations were detected with Cobas^®^ and 42 with UltraSEEK™, with 7 additional mutations in e.g., *BRAF* and *KRAS* not present on the Cobas^®^ Panel, leading to an agreement of 0.49 (Figure 4C,D; Appendix A).

Using (sub-)optimal ccfDNA input (>8 ng), a 100% concordance was observed (Table 4). However, with lower input amounts the concordance drops to 76% and 73% for 2–8 ng and Liquid IQ^®^ unevaluable samples, respectively. After exclusion of mutation-negative (wildtype) plasma samples (*n* = 26), still a 100% concordance was observed when using >8 ng of ccfDNA input (Table 4; Appendix A). Interestingly, a substantial drop in concordance is observed in samples using 2–8 ng of ccfDNA (54%), and even no concordance in samples with unevaluable ccfDNA input (Table 4; Appendix A). This emphasizes the necessity of analyzing preanalytical sample conditions prior to molecular mutation profiling to ensure accurate variant calling and treatment decision making.

## 3. Discussion

Liquid biopsy is a promising approach for noninvasive assessment of cancer gene profiles and, although it is currently solely used as an alternative when tissue is not available, it can be used to identify mutations occurring in lung cancer both at diagnosis and during the course of disease. Liquid biopsy approaches are increasingly applied in the clinical setting due to convenience and improved test reliability and accuracy [11,26,27]. Unlike traditional tumor tissue biopsies, ccfDNA analysis harbors the potential to detect and monitor the consistently evolving mutational profile of early stage and (oligo-) metastasized cancer. In this regard, performance characterizations of current methods, especially sensitivity and specificity and preanalytical conditions, are of major importance. 

Here, we compared the performance of the Cobas^®^ EGFR v2 test with the novel UltraSEEK™ Lung Panel on the MassARRAY^®^ System on routine clinical samples of NSCLC patients. An overall concordance of 86% was observed, with a substantial agreement (κ = 0.71), of those *EGFR* mutations covered by both assays when assay performance was according to the manufacturer’s instructions. The sensitivity of the Cobas^®^ and UltraSEEK™ tests were similar and an almost equal number of mutations were missed. For primary testing, *EGFR* exon19 deletions and L858R mutations are key to initiate EGFR-TKI treatment. UltraSEEK™ analysis revealed a lower sensitivity than the Cobas^®^ test in respect of detecting these mutations. The most common resistance mutation upon gefitinib/erlotinib treatment *EGFR* T790M is found in over 50% of cases and these patients are eligible for treatment with osimertinib [28]. The detection of *EGFR* T790M mutations in cases with progression on TKI is relevant to alter treatment strategies, for which the Cobas^®^ test in plasma at progression is FDA-approved. However, 6 out of 34 cases with an *EGFR* T790M mutation were missed when performing Cobas^®^ testing, while all were detected with UltraSEEK™ analysis. Of the 77 patients of which tissue sequencing data was available, 92% of the mutations were detected in both tissue and plasma (Appendix A). In total, 19 mutations were only detected in plasma, especially therapy-induced *EGFR* T790M mutations (*n* = 17), while 7 tissue mutations were missed in plasma, emphasizing the importance of sensitive plasma-based monitoring of the mutational profile for accurate treatment decision making.

Regarding all genes included in the panels, UltraSEEK™ analysis revealed more clinically relevant mutations in almost a quarter of the studied patients. Besides *EGFR* T790M, other resistance mutations can only be detected by the UltraSEEK™ Lung Panel (in *BRAF*, *ERBB2*, *KRAS* and *PIK3CA*, as well as *EGFR* C797S) and could be are useful for treatment decision making for lung cancer patients. Since osimertinib is now first-line TKI in activating exon 19 deletions and L858R EGFR mutations, the resistant mutation profiles will be changing with much less T790M occurrence. Therefore, a broader range of new resistance mutations, such as *EGFR* C797S or mutations on non-*EGFR* genes should be incorporated into new assays. 

As preanalytical conditions, DNA extraction quality and quantity affect mutation detection [14]. Here, we introduce the Liquid IQ^®^ Panel to determine ccfDNA quality and quantity using the MassARRAY^®^ System. The accuracy of the detection method is not only dependent on the tumor-to-wildtype DNA ratio in cell-free plasma as determined by the VAF, but also by the total amount of ccfDNA input. We have recently reported that decreasing DNA input lowers the accuracy of VAF detection using next-generation sequencing, droplet digital PCR (ddPCR) as well as the UltraSEEK™ approaches [15]. The Liquid IQ^®^ Panel determines DNA quantity (determination of the number of amplifiable ccfDNA copies) and DNA quality (identification of long DNA fragments (>340 bp) from cell necrosis and WBC contamination). In terms of quantification, we showed a fair concordance between the Liquid IQ^®^ Panel, the fluorescence quantification by Qubit™ and the microfluidic electrophoresis performed on a LabChip^®^. Both the LabChip^®^ and Qubit™ detected overall amounts of ccfDNA, whereas Liquid IQ^®^ analysis only detected amplifiable ccfDNA. Qubit™ seems to seriously overestimate the quantity of ccfDNA available for amplification, which is in agreement with previous reports comparing Qubit™ and PCR-based quantitative analyses [14].

Using the Liquid IQ^®^ Panel, a significant contribution of the ccfDNA input amount to the accuracy of mutation detected was observed. When using (sub-)optimal input amounts (>8 ng ccfDNA), we showed a concordance of >94% between the Cobas^®^ and UltraSEEK™ tests, which is even 100% when using >10 ng ccfDNA. Furthermore, all tissue confirmed mutations were detected in plasma (Appendix A). If less ccfDNA input was used, the concordance dropped to 73–84%, implicating a loss of sensitivity. When solely considering mutation positive samples, the concordance is even lower for the lesser input amounts (67–69%) while it is similar in (sub-)optimal conditions (93%). However, in many cases patient-derived samples do not accommodate for such high ccfDNA input amounts. To determine the concordance between Cobas^®^ and UltraSEEK™ with similarly high or low ccfDNA input amounts, harmonized eluate volumes were used, which is not according to the FDA-approved recommendations of the Cobas^®^ test. The concordance of detecting *EGFR* mutations present in both panels was 78% (κ = 0.66). The UltraSEEK™ assay detected 14% more mutations while the Cobas^®^ test detected only 2% more. Four of the Cobas^®^ analyses failed, probably due to insufficient input. More importantly, the concordance was again impaired when using <8 ng of ccfDNA (64–76%) in contrast to a 100% concordance with (sub-)optimal input. Particularly when excluding wildtype samples, only half the cases were concordant using 2–5 ng ccfDNA and even none in unevaluable material. To conclude, for reliable mutation detection with UltraSEEK™, a minimal of 10 ng ccfDNA input is required. In cases at lower input amounts, the presence of a mutation cannot be excluded in negative results. 

This analysis underlines the importance of reliable ccfDNA quantification through collection of the actual number of ccfDNA amplifiable copies to ensure sufficient input. The UltraSEEK™ Lung Panel shows promising results to accurately call clinically relevant mutations when preanalytical conditions are met. The major challenge regarding ctDNA analysis is the release of genomic DNA from leukocytes during sample storage and preparation. Total ccfDNA could be elevated due to inaccurate preanalytical procedures leading to false negative results [29]. As liquid biopsy approaches are considered to be applied for disease monitoring, hemolysis could dramatically affect the quantification of VAFs, resulting in imprecise clinical interpretation thereof. The Liquid IQ^®^ Panel is able to detect WBC-derived DNA contamination in samples. Liquid IQ^®^ analysis on the 137 plasma samples revealed twelve samples with WBC contamination of >10%. Those samples were more likely to present discordant results between tissue DNA and parallel ccfDNA testing (Appendix A). These have been critically analyzed with regard to mutation detection. Interestingly, solely a single case with >10% WBC contamination showed discordant results in detecting clinically relevant mutations. At least, WBC-derived ccfDNA potentially carries non-cancerous mutated DNA. Recent studies found that clonal hematopoiesis of indeterminate potential, (CHIP) can significantly contribute to the detection of false-positive mutations on genes like *KRAS* [30,31,32]. 

## 4. Materials and Methods 

### 4.1. Patient Inclusion and Plasma Sample Processing

Cell-free plasma samples of patients with advanced NSCLC previously tested with the Cobas^®^ EGFR Mutation Test v2 in the routine diagnostic setting (Figure 5) were initially selected from the Montpellier biobank for technique optimization and ISO15189 accreditation purposes (training series; *n* = 31). Afterwards, similar samples were retrieved from the certified S96-900 LCPE biobank of Nice (MTP cohort; *n* = 100) and Groningen (GRO cohort; *n* = 37) biobanks, authorized by the medical ethical committee (METc, 2010/109) of the university medical center Groningen (UMCG). These patients were referred for ctDNA *EGFR* molecular analysis because of disease progression on EGFR-TKI or because no tissue biopsy could be obtained at first presentation with NSCLC. The Cobas^®^ EGFR V2 test analysis 46 EGFR mutation at DNA level was performed following the company’s instructions. For both the MTP and GRO cohorts, molecular testing for Cobas^®^ were performed in the ISO15189-accredited laboratory of molecular pathology. Cobas^®^ test results were retrieved from the pathological molecular diagnostics archives or biobank database.

Plasma samples were collected in cell-free DNA blood collection tubes (BCTs) (Streck, Omaha, NE, USA), except for 33 samples collected in EDTA tubes in the MTP cohort. Cell-free DNA BCTs were processed within 24 hours and EDTA samples within 4 hours after blood collection. Samples were centrifuged at 1600× *g* (Streck and EDTA) for 10 minutes to separate lymphocytes from plasma. The supernatant was subsequently centrifuged for 10 minutes at 16,000× *g* to remove the remaining debris. Cell-free plasma was stored in 1 mL fractions at −80 °C until ccfDNA extraction. For the validation of the Liquid IQ^®^ Panel, from the same 42 patients simultaneously plasma was collected in a Streck (*n* = 42) and EDTA (*n* = 42) blood collection tubes. Samples were randomly selected from routine sample collection for storage in the UMCG biobank during a small period of time. All patients gave written informed consent.

### 4.2. CcfDNA Extraction and Sample Quantity and Quality Assessment

For the validation of ctDNA using UltraSEEK™ and ddPCR analyses, ccfDNA was extracted from 4 mL (MTP cohort) and 2 mL (GRO cohort) of the same cell-free plasma used for the diagnostic Cobas test and eluted in 100 µL (MTP cohort) and 52 µL (GRO cohort) of elution buffer using the QIAamp Circulating Nucleic Acid Kit (Qiagen, Hilden, Germany) according to the manufacturer’s recommendations as reported previously [16,17,18,19,20,21,22,23,24,25,26,27,28,29,30,31,32,33]. CcfDNA yield was determined with either the LabChip^®^ GX Touch™ Nucleic Acid Analyzer (PerkinElmer, Waltham, MA, USA; MTP cohort) or the Qubit™ 1× dsDNA HS Assay Kit (Thermofisher Scientific, Waltham, MA, USA; GRO cohort). 

Single-reaction assessment of multiple preanalytical parameters was performed using 1.5 µL extracted ccfDNA from each sample (MTP cohort singular or duplicate analysis; GRO cohort in triplicate) with the novel Liquid IQ^®^ Panel with MALDI-TOF-based analysis on the MassARRAY^®^ System System (Agena Bioscience). Briefly, this preanalytical control can detect long DNA fragments (>340 bp) originating from cell necrosis and WBC contamination, based on the number of amplifiable ccfDNA copies, to give an estimation of the level of WBC contamination. Taking these results into account, the Liquid IQ^®^ Panel analysis calculates the optimal ccfDNA input. In addition, analysis with the Liquid IQ^®^ Panel also provides sample tracking information, matching liquid and tissue biopsy samples across studies and prevention of potential sample mix-up with genetic barcodes from 21 SNPs and gender markers.

### 4.3. Molecular Analysis 

The UltraSEEK™ Lung Panel is a liquid biopsy test performed on the MassARRAY^®^ System (Agena Bioscience) with a specific mutant allele capture that increase signal for mutation detection. It was designed to detect 73 variants at DNA level across 5 genes relevant to NSCLC (46 *EGFR*, 15 *KRAS*, 4 *BRAF*, 4 *ERBB2* and 4 *PIK3CA*; see Appendix A). An initial training series for UltraSEEK Lung Panel was performed in Montpellier leading to method adjustments, routine implantation and ISO15189 accreditation [34]. Sensitivity, specificity, interlaboratory reproducibility and analytical limit of detection of the kit were determined in a ring trail. The limit of detection was 0.125% for the *EGFR* exon19 deletions to 1% for *EGFR* exon20 insertions [35]. 

Then, the UltraSEEK™ analysis was performed on ccfDNA of all 137 plasma samples of the MTP and GRO cohorts. For the MTP cohort, a fixed input volume of 40 µL of the ccfDNA eluate was applied, except when less material was available (*n* = 24). For the GRO cohort, ccfDNA input was ideally 10 ng as determined with the Liquid IQ^®^ Panel on the MassARRAY^®^ System. Since this input was not always achievable for all samples due to the limited input volume per reaction, ccfDNA input median was 6.8 ng (3–25 ng) for the MTP cohort and 9.4 ng (2–10 ng) for the GRO cohort. A set of controls (MTP cohort: 0.5% VAF EGFR Multiplex cfDNA Reference Standard and EGFR Multiplex Wild-Type control, Horizon Discovery, Waterbeach, UK; GRO cohort: 0.5% VAF SeraSeq ctDNA Complete reference material and SeraSeq Complete WT control, SeraCare Life Sciences Inc, Milford, MA, USA) were added to each run. 

In addition, harmonized ccfDNA inputs for 63 samples from the MTP cohort were reanalyzed using the Cobas^®^ test on the same eluate as applied for the UltraSEEK™ analysis. To this extend, a similar input volume from the same eluate as for the UltraSEEK™ analysis was used for the Cobas^®^ test enabling direct comparison of mutation detection results. Except for the ccfDNA extraction procedure, the assay was performed according to manufacturer’s instructions.

### 4.4. Statistical Analyses 

Statistical analyses were performed using Prism 8.4.2. (GraphPad software, San Diego, CA, USA). Agreement of mutation detection between UltraSEEK™ and Cobas^®^ analyses was performed using Cohen’s κ. Correlations were determined using Pearson’s correlation coefficient. For statistical assessment between two groups, a Wilcoxon matched-pairs signed rank test was applied.

## 5. Conclusions

This study showed that the UltraSEEK™ Lung Panel on the MassARRAY^®^ System is a sensitive and accurate technology for clinically relevant mutation detection using ccfDNA. The UltraSEEK™ method showed similar sensitivity in detection of *EGFR* mutations compared to the FDA-approved Cobas^®^. Particularly in patients with low quantity of ccfDNA, the detection of false-negative cases increases significantly. It is required to determine the ccfDNA concentration accurately to ensure sufficient input for reliable interpretation. For the detection of relevant mutations in liquid biopsy in NSCLC using the UltraSEEK™, an input of over 10 ng showed 100% concordance with Cobas^®^ and detection of all tissue confirmed mutations. In case of lower ccfDNA input, the risk of missing clinically relevant mutations should be considered. The use of a preanalytical ccfDNA quality control assay such as the Liquid IQ^®^ Panel on the MassARRAY^®^ System is recommended to confidently interpret results, avoiding bias induced by non-specific genomic DNA or low input of specific tumoral ccfDNA fragments. The use of a multigene panel covering clinical actionable and resistance mutations is important to get a full mutational status profile of a patient as a guidance for therapy. 

## Figures and Tables

**Figure 1 cancers-12-03002-f001:**
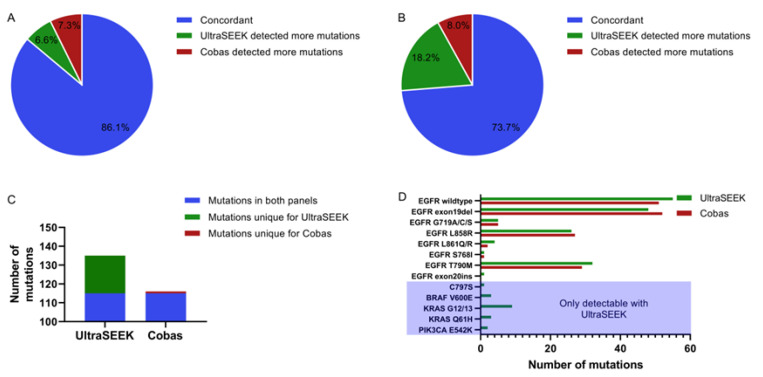
Comparison of mutation detection between UltraSEEK™ and Cobas^®^. Pie charts representing the samples which were concordant between UltraSEEK™ and Cobas^®^ (blue), samples in which UltraSEEK™ detected more mutations (green), and samples in which Cobas^®^ detected more mutations (red) for (**A**) the mutations detectable on both panels and (**B**) all clinically relevant mutations across *BRAF*, *EGFR*, *KRAS*, *ERBB2* and *PIK3CA*. (**C**) Bar graph illustrating the total amount of mutations detected in all 137 samples. The blue bar represents the number of mutations that could be detected on both panels; the green bar represents detected mutations unique for the UltraSEEK™ Panel. One *EGFR* exon19 deletion genotype was detected with Cobas^®^ that is not represented on the UltraSEEK™ Panel (red bar). (**D**) Bar graph illustrating frequency of detection per mutation with UltraSEEK™ (green) and Cobas (red).

**Figure 2 cancers-12-03002-f002:**
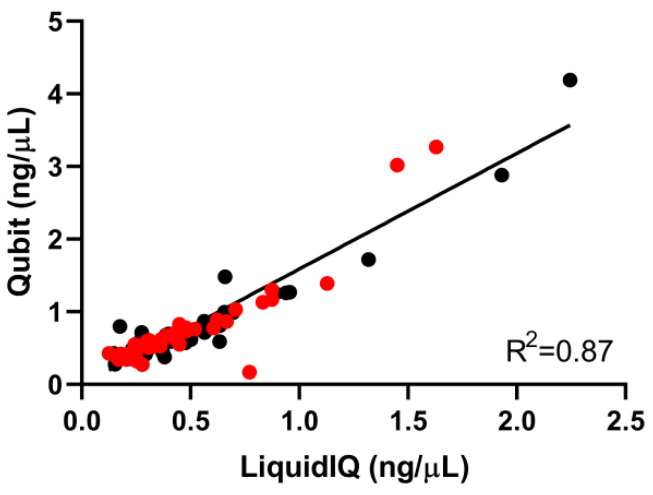
Correlation between Liquid IQ^®^ and Qubit™ results of Streck and EDTA plasma samples. The correlation between Liquid IQ^®^ and Qubit™ for the ccfDNA extracted from EDTA (black; R^2^ = 0.91) and Streck (red; R^2^ = 0.82) separately showed similar results.

**Figure 3 cancers-12-03002-f003:**
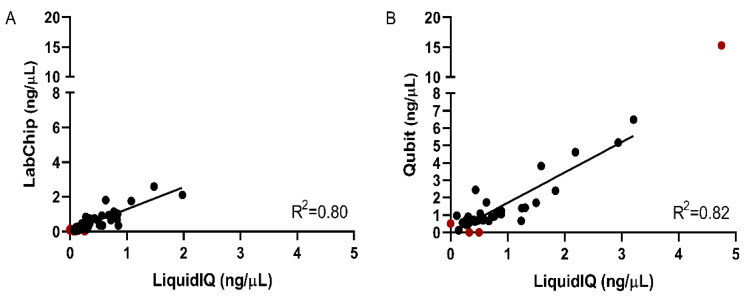
Correlation of Liquid IQ^®^ and a secondary DNA quantification assay. Correlation of the ccfDNA concentration as determined by the amplifiable copies measured with Liquid IQ^®^ and the quantification based on (**A**) LabChip^®^ performed on the samples of the MTP cohort and (**B**) Qubit™ performed on the samples of the GRO cohort. Red-colored results are not included in the correlation due to failure of either quantification assay or was considered an outlier.

**Figure 4 cancers-12-03002-f004:**
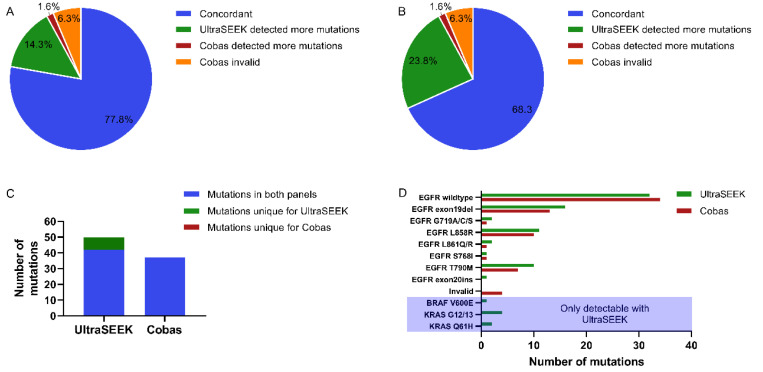
Comparison of mutation detection between UltraSEEK™ and Cobas^®^ with harmonized sample input. Pie charts representing the samples which were concordant between UltraSEEK™ and Cobas^®^ (blue), samples in which UltraSEEK™ detected more mutations (green), samples in which Cobas^®^ detected more mutations (red), and sample of which the Cobas^®^ analysis was invalid supposedly due to insufficient input (purple) for (**A**) the mutations detectable on both panels and (**B**) all clinically relevant mutations. (**C**) Bar graph illustrating the total amount of mutations detected in all samples. The blue bar represents the number of mutations that could be detected on both panels; the green bar represents detected mutations unique for the UltraSEEK™ Panel. No mutations only covered by the Cobas^®^ Panel were detected (red bar). (**D**) Bar graph illustrating frequency of detection per mutation with UltraSEEK™ (green) and Cobas (red).

**Figure 5 cancers-12-03002-f005:**
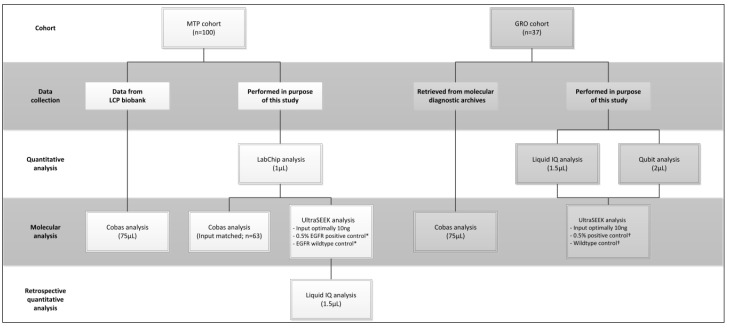
Flowchart of data collection. * Reference material from Horizon Discovery, Waterbeach, UK. † Reference material from SeraCare Life Sciences Inc, Milford, MA, USA.

**Table 1 cancers-12-03002-t001:** Liquid IQ^®^ results for the Streck and EDTA samples.

Tube Type	SNP Calls	Amplifiable Copies	WBC Contamination	Yield (ng/µL)	Qubit (ng/µL)	No Result *
Streck (*n* = 42)	20 (16–21)	179 (57–734)	3% (0–28%)	0.40 (0.13–1.6)	0.62 (0.43–3.3)	0
EDTA (*n* = 42)	20 (15–21)	177 (66–1010)	1% (0–35%)	0.39 (0.15–2.3)	0.63 (0.49–4.2)	0

Data are presented as median with range between brackets. * Level of amplifiable copies too low for accurate calling and considered unevaluable.

**Table 2 cancers-12-03002-t002:** Liquid IQ^®^ results for the Montpellier (MTP) and Groningen (GRO) cohorts.

Patient’s Cohort	SNP Calls	Amplifiable Copies	WBC Contamination	Yield (ng/µL)	Input (ng)	No Result *
MTP cohort (*n* = 100)	20 (15–21)	86 (30–891)	0% (0–74%)	0.19 (0.067–2.0)	6.8 (2.7–25)	24
GRO cohort (*n* = 37)	20 (15–21)	257 (50–2136)	1% (0–44%)	0.57 (0.11–4.8)	9.4 (2.3–10)	1

Data are presented as median with range between brackets. * Level of amplifiable copies too low for accurate calling and considered unevaluable.

**Table 3 cancers-12-03002-t003:** Concordance of mutation detection between UltraSEEK™ and Cobas^®^ according to ccfDNA input amount.

Input *	Cases	Concordant	Cobas Detected More Mutations	UltraSEEK Detected More Mutations	Grouped Concordance	Excluding Wildtype ^‡^
Unevaluable ^†^	25	21 (84%)	1 (4%)	3 (12%)	21 (84%)	8/12 (67%)
2–5 ng	32	29 (91%)	2 (6%)	1 (3%)	46 (79%)	27/39 (69%)
5–8 ng	26	17 (65%)	5 (19%)	4 (15%)		
8–10 ng	31	28 (90%)	2 (6%)	1 (3%)	51 (94%)	37/40 (93%)
>10 ng	23	23 (100%)	0 (0%)	0 (0%)		

* as determined by Liquid IQ^®^ analysis. ^†^ Level of amplifiable copies too low for accurate calling and considered unevaluable. ^‡^ Mutation-negative samples are considered wildtype samples, see Appendix A.

**Table 4 cancers-12-03002-t004:** Concordance of mutation detection related to the harmonized ccfDNA input.

Input *	Cases	Concordant	Cobas Detected More Mutations	UltraSEEK Detected More Mutations	Grouped Concordance	Excluding Wildtype ^‡^
Unevaluable ^†^	15	11 (73%)	1 (7%)	3 (20%)	11 (73%)	0/4 (0%)
2–5 ng	15	13 (87%)	0 (0%)	2 (13%)	19 (76%)	7/13 (54%)
5–8 ng	10	6 (60%)	0 (0%)	4 (40%)		
8–10 ng	7	7 (100%)	0 (0%)	0 (0%)	23 (100%)	16/16 (100%)
>10 ng	16	16 (100%)	0 (0%)	0 (0%)		

Data are excluding the four cases of which the Cobas^®^ test was invalid. * as determined by Liquid IQ^®^ analysis. ^†^ Level of amplifiable copies too low for accurate calling and considered negative. ^‡^ Mutation-negative samples are considered wildtype samples, see Appendix A.

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
