# Peer review of "Mass Spectrometry as a Highly Sensitive Method for Specific Circulating Tumor DNA Analysis in NSCLC: A Comparison Study"

_cancers, 2020, doi:10.3390/cancers12103002_

Round 1
Reviewer 1 Report
In this study by Lamy et al., the authors present a molecular and clinically relevant study examining the usefulness of the Mass Spectrophotometry-based method, UltraSeek, with the FDA-approved EGFR Cobas mutation v2 assay in 132 plasma samples from NSCLC patients using two cohorts from Montpellier (MTP, France) and Groningen (GRO, Netherlands).
In light of the emerging interest and use of the liquid biopsy in lung cancer where tumour tissue may not be accessible or insufficient tumour material obtained from biospy, this study is of importance to both clinicians and scientists in mutation profiling of NSCLC patient material. Of particular importance in the current study, are the data relating to the input of ccfDNA from blood plasma with a minimum requirement of 10ng to ensure optimal mutation detection and a reduced false-negative result.
There are a number of minor questions highlighted below based on the current content of this manuscript which should be addressed, where possible, by the authors:
(1) In Table 2, showing data for the Liquid IQ between the two cohorts (MTP and GRO), despite the smaller GRO cohort, there is clearly a substantially larger number of amplifiable copies detected relative to those identified for the MTP cohort (257 vs 86, respectively). Based on the clinical and/or pathological features of patients within this cohort, is there any evidence to suggest this larger increase in numbers?
(2) Figure 3 represents data showing correlations of the Liquid IQ and secondary DNA quantification assays, namely LabChip and Qubit. However, the comparisons used are different between the two cohorts (MTP & GRO), where a Liquid IQ vs LabChip comparison is used for the MTP cohort, while a Liquid IQ vs Qubit comparison is made for the GRO cohort. Are data available showing both comparisons across both MTP and GRO cohorts?
(3) Results section, 2.4, the authors refer to Table S1 for ccfDNA inputs determined by Liquid IQ analysis. Based on the data available, should this not be Table S2?
(4) Throughout the results section of the manuscripts, the authors refer to the pie-charts as "circle diagrams". This should be amended throughout to the more appropriate terminology of "pie-chart".
(5) Materials & Methods, 4.1. If ethical approval was granted for this study, the local IRB who reviewed and approved this, should be included in this section, together with a reference number.
(6) Reference [36] does not appear to be referred to in the main body of text.
(7) There are some minor typos and spelling errors within the manuscript that should be checked and corrected.
Author Response
Dear Editor,
We would like to sincerely thanks the reviewer for their kind comments, the wise questions they address and their useful recommendations. Please find below our point by point response to reviewer. All relevant modifications were added in the revised manuscript.
Sincerely,
On behalf the authors,
Dr Pierre-Jean Lamy
Reply to reviewer 1.
(1) In Table 2, showing data for the Liquid IQ between the two cohorts (MTP and GRO), despite the smaller GRO cohort, there is clearly a substantially larger number of amplifiable copies detected relative to those identified for the MTP cohort (257 vs 86, respectively). Based on the clinical and/or pathological features of patients within this cohort, is there any evidence to suggest this larger increase in numbers?
Thank you for underlying this point. Although clinical and pathological features are unavailable for many of the included patients, it is known that all included patients have advanced metastatic disease (stage IIIB or higher). Based on other clinical features available, no obvious distinctions could be observed between the two cohorts and, as stated in the manuscript, no detailed tumor tissue data is available in the majority of patients. Besides, we specified that MPL cohort used 4mL of plasma for DNA extraction while the GRO cohort used 2 mL of plasma. Even though alternate elution volume (100 and 52 µL respectively) were standardized, this could explanation the observed variation. Besides, there is possible another explanation due to preanalytical conditions like type of tubes (33 were in EDTA tube for the MPL cohort) or slight variation on time to centrifugation. However, considering that our study is a retrospective one, it is difficult to conclude what the exact reason is for this observation.
(2) Figure 3 represents data showing correlations of the Liquid IQ and secondary DNA quantification assays, namely LabChip and Qubit. However, the comparisons used are different between the two cohorts (MTP & GRO), where a Liquid IQ vs LabChip comparison is used for the MTP cohort, while a Liquid IQ vs Qubit comparison is made for the GRO cohort. Are data available showing both comparisons across both MTP and GRO cohorts?
All samples were characterized with Liquid IQ. Unfortunately, we did not use a similar secondary quantitative assay across both cohorts. Since this is a retrospective study, this was not harmonized. When the data for the Liquid IQ vs Qubit and Liquid IQ vs LabChip are combine, you can appreciate a strong correlation between the Liquid IQ and the secondary quantification assays (see below, data not shown in the manuscript). However, in the manuscript we would like to display the slight difference in the angle of the curves between Qubit and LabChip, which shows that Qubit overestimates the amount of DNA in the eluates. Therefore, Figure 3 is presented as two separate graphs.
(3) Results section, 2.4, the authors refer to Table S1 for ccfDNA inputs determined by Liquid IQ analysis. Based on the data available, should this not be Table S2?
We thank the reviewer for noticing this mistake. The text has been altered to Table S2.
(4) Throughout the results section of the manuscripts, the authors refer to the pie-charts as "circle diagrams". This should be amended throughout to the more appropriate terminology of "pie-chart".
This suggestion has been adopted in the revised manuscript (legend Figure 1; legend Figure 4)
(5) Materials & Methods, 4.1. If ethical approval was granted for this study, the local IRB who reviewed and approved this, should be included in this section, together with a reference number.
According to the reviewer’s suggestion, we added information regarding the ethical authorization of this study to section 4.1.
(6) Reference [36] does not appear to be referred to in the main body of text.
We thank the reviewer for noticing. The minor mistake in the references section has been resolved.
(7) There are some minor typos and spelling errors within the manuscript that should be checked and corrected.
The manuscript has been revised thoroughly for the English writing style, including one of the co-authors who has lived in the USA and Canada for many years.

Reviewer 2 Report
In this work, Lamy and colleagues excellently compared the performance of the Cobas® EGFR v2 test with the novel UltraSEEK™ Lung Panel on the MassARRAY® System on a cohort of NSCLCs (n=137 patient-derived cell-free plasma samples) with an overall concordance of 86% of EGFR-mutations covered by both assays. Looking at the EGFR T790M mutations, they showed the utility of UltraSEEK™ analysis detecting all of these, differently from Cobas® (in 6/34 cases were not identified). Moreover, using the Liquid IQ® Panel, they realized a significant contribution of the ccfDNA input amount to the accuracy of mutation detected. In this regard, they showed (when using a >8ng of input amounts) a ccfDNA concordance of >94% between the Cobas® and UltraSEEK™ tests, which even is 100% when using >10ng ccfDNA. They concluded by saying that these results emphasized the importance of sensitive plasma-based monitoring of the mutational profile for accurate treatment decision making in NSCLC patients’.
To my eyes, the paper is a great technical paper by considering the novelty of the aim and the methodology used for data analysis and the results are interesting. The scientific content seems good as well as the English style and language used. Moreover, I appreciate the scientific efforts to organize this paper and I think that the rationale is well-argued. Firstly, the methods compared seem adequate and quite-structured. Secondly, the results are intriguing and well-organized in each section. The resolution of the image/graphic of each experiment meets the quality requirements of our journal. Moreover, the results were clearly discussed and corroborated with what is shown. Finally, the data analyses were interpreted in a comprehensible manner as well as statistical tests applied. I didn’t observe any remarkable incongruences throughout the text. Overall, I agree with the authors in terms of methodologies, results and comments of data, supporting the proposal of their considerable work.
I have only two questions/clarifications before considering the manuscript accepted.
- Firstly, I didn’t see any clinical pathological data about NSCLC patients. I kindly ask you to report them related to the disease staging. Taking into account that the UltraSEEK technology can detect mutations in cfDNA in plasma with oligo-metastatic tumor burden in NSCLC patients’, I wonder if this could be useful for a non-invasive stratification of tumour heterogeneity for personalized cancer therapy in NSCLC patient populations with different outcomes. Please clarify it.
- In a technical context, you worked with blood EDTA samples. I would like to know the exact volume that you used to extract plasma. Please add it.
Author Response
Dear Editor,
We would like to sincerely thanks the reviewers for their kind comments, the wise questions they addressed and their useful recommendations. Please find below our point by point response to reviewer. All relevant modifications were added in the revised manuscript.
Sincerely,
On behalf the authors,
Dr Pierre-Jean Lamy
Reply to reviewer 2.
Firstly, I didn’t see any clinical pathological data about NSCLC patients. I kindly ask you to report them related to the disease staging. Taking into account that the UltraSEEK technology can detect mutations in cfDNA in plasma with oligo-metastatic tumor burden in NSCLC patients’, I wonder if this could be useful for a non-invasive stratification of tumour heterogeneity for personalized cancer therapy in NSCLC patient populations with different outcomes. Please clarify it.
Samples for this study were taken from patients during their treatment. CtDNA analyses were prescribed in order to treat those patients with RTKI or to detect resistance mutation after initial RTKI treatment. At this time, only patients with an advance disease (stages IIIB-IV) were eligible for RTKI of 1st, 2nd or 3rd generation. Therefore, we cannot show data for patients with early disease. It is beyond the scope of this technical evaluation manuscript to extensively investigate the mutational profile of the patients, primarily since tumor tissue data is unavailable for a substantial part of the patient. We did, however, detect several patients with 2 or 3 mutations possibly related with tumor heterogeneity. Naturally, the potential of ccfDNA analysis of (oligo-)metastatic cancer is highly relevant. A notion of this has been added to the discussion.
- In a technical context, you worked with blood EDTA samples. I would like to know the exact volume that you used to extract plasma. Please add it.
Although most of the samples were collected in Streck BCTs, indeed a part of the MTP samples were collected in EDTA BCTs. Of the samples collected in EDTA BCTs, 4 mL of plasma was used for the ccfDNA extraction, similar as for the Streck BCTs in the MTP cohort. In the material and method section, volume of plasma used for DNA extraction was precised: L 350-354 :“ccfDNA was extracted from 4mL (MTP cohort) and 2mL (GRO cohort) of the same cell-free plasma used for the diagnostic Cobas test and eluted in 100μL (MTP cohort) and 52μL (GRO cohort) of elution buffer using the QIAamp Circulating Nucleic Acid Kit (Qiagen, Hilden, Germany) according to the manufacturer's recommendations as reported previously”
